# Obesity as a Risk Factor for Severe COVID-19 and Complications: A Review

**DOI:** 10.3390/cells10040933

**Published:** 2021-04-17

**Authors:** Fien Demeulemeester, Karin de Punder, Marloes van Heijningen, Femke van Doesburg

**Affiliations:** Natura Foundation, 3281 NC Numansdorp, The Netherlands; k.depunder@naturafoundation.com (K.d.P.); m.vanheijningen@naturafoundation.com (M.v.H.); f.vandoesburg@naturafoundation.com (F.v.D.)

**Keywords:** coagulopathy, COVID-19, cytokine storm, inflammation, leptin, obesity, SARS-CoV-2

## Abstract

Emerging data suggest that obesity is a major risk factor for the progression of major complications such as acute respiratory distress syndrome (ARDS), cytokine storm and coagulopathy in COVID-19. Understanding the mechanisms underlying the link between obesity and disease severity as a result of SARS-CoV-2 infection is crucial for the development of new therapeutic interventions and preventive measures in this high-risk group. We propose that multiple features of obesity contribute to the prevalence of severe COVID-19 and complications. First, viral entry can be facilitated by the upregulation of viral entry receptors, like angiotensin-converting enzyme 2 (ACE2), among others. Second, obesity-induced chronic inflammation and disruptions of insulin and leptin signaling can result in impaired viral clearance and a disproportionate or hyper-inflammatory response, which together with elevated ferritin levels can be a direct cause for ARDS and cytokine storm. Third, the negative consequences of obesity on blood coagulation can contribute to the progression of thrombus formation and hemorrhage. In this review we first summarize clinical findings on the relationship between obesity and COVID-19 disease severity and then further discuss potential mechanisms that could explain the risk for major complications in patients suffering from obesity.

## 1. Introduction

The novel coronavirus SARS-CoV-2 that first appeared in the Chinese city of Wuhan is still causing a global pandemic with a death toll that already exceeds 2,430,000 people and continues to grow every day (number from John Hopkins University). SARS-CoV-2 is the virus that causes COVID-19, a clinical picture characterized by fever, coughing, muscle pain and fatigue and can evolve into hyperinflammation, cytokine storm, ARDS and COVID-associated-coagulopathy (CAC) [1,2]. A large number of patients severely ill with COVID-19 arriving at the ICU are overweight or suffer from obesity [3]. These conditions, as well as smoking, age, type II diabetes and cardiovascular diseases, appear to be major risk factors for serious complications and increased mortality in COVID-19 patients [4,5,6].

Understanding the contributing factors to COVID-19 disease severity and complications in the context of obesity is of key importance for the development of therapeutic interventions, as well as for advancing preventative strategies in this high-risk group. Therefore, in this review we first summarize clinical findings on obesity and COVID-19 disease outcomes. Next, we discuss possible underlying mechanisms linking obesity to major disease complications as a result of SARS-CoV-2 infection Here we focus on the metabolic- and immune-related consequences of obesity on COVID-19 disease course.

## 2. Obesity

### 2.1. Obesity Is a Common Disease Associated with Chronic Inflammation and Insulin and Leptin Resistance

The prevalence of obesity has increased worldwide over the last 50 years. In 2015 the mean prevalence of obesity in adults of selected countries was 19.5% and ranged from 3.7% in Japan to 38.2% in the United States [7]. Obesity (BMI ≥ 30 kg/m^2^) is a major risk factor for the development of non-communicable diseases and is defined by the WHO as abnormal or excessive fat accumulation that might impair health [8].

Fat or adipose tissue, originally regarded as a simple organ for storing energy, is currently viewed as one of the most important endocrine organs [9,10,11]. Fat cells, or adipocytes, produce cytokine-like hormones, called adipokines, which play a major role in metabolism and inflammation [11]. Adipocytes in different regions fulfill different functions. Distinctions should be made between brown and white adipocytes, between ectopic and non-ectopic and between visceral and subcutaneous adipose tissue to accurately estimate the impact of the adipose tissue on the patient’s health. For example, visceral fat, but not so much subcutaneous fat accumulation, promotes systemic inflammation and is associated with impairments of glucose and lipid metabolism accompanied by insulin resistance [12,13,14].

Obesity is associated with chronic inflammation, resulting from immune cell activity in dysfunctional (visceral) adipose tissue. In obesity, the excessive presence and hypertrophy of adipocytes result in hypoxia, cell stress and apoptosis. The hypoxic environment induces the infiltration of immune cells into the adipose tissue as a result of the expression of chemo attractive molecules [15]. Also, hypertrophic adipocytes produce multiple pro-inflammatory adipokines, such as such as tumor necrosis factor-α (TNF-α), interleukin (IL)-6 and leptin [16].

Leptin informs the brain about the amount of energy stored in the adipose tissue [17,18]. The percentage of total body fat is the most important determinant of leptin levels [19]. Given their high percentage of total body fat, individuals suffering from obesity show elevated leptin concentrations [20], a condition also referred to as hyperleptinemia. Persistent hyperleptinemia is often accompanied with central leptin resistance, due to which appetite and satiety regulation is disrupted [21]. In addition to the regulation of hunger and satiety, leptin also has pro-inflammatory properties [22], further contributing to a chronic inflammatory state in individuals suffering from obesity [23].

### 2.2. Obesity and COVID-19 Disease Severity

In the following section, we summarize clinical findings supporting an association between obesity and COVID-19 disease severity (see also Table 1). We searched the Pubmed database up to March 15, 2021. In our search strategy, the combination of the following keywords was used: obesity OR BMI OR overweight OR adiposity OR adipose tissue AND COVID-19 OR SARS-CoV-2. Our search was limited by published full-text article in English language. Most studies investigating this association used BMI categories as the predictor variable [24,25,26,27,28,29,30,31,32,33,34,35,36,37]. Four cross-sectional studies did not specify the used classification of obesity [38,39,40,41] presumably using the WHO guidelines defining obesity as BMI of 30 or higher. Outcome variables included: hospitalization [36,38,42], ICU admission [31,35,37,38,43,44,45], intubation [24,25,29,37,38], invasive mechanical ventilation [26,31,34], disease severity [27,28,30,33] and death [24,25,26,27,28,29,32,38,39,40,41]. Of the eleven studies investigating the association between BMI and mortality in hospitalized COVID-19 patients, ten studies observed an increased mortality rate in patients that were overweight (BMI ≥ 25 to <30) [24,32], or suffering from obesity (BMI ≥ 30) [25,29,38,39,40,41], or severe obesity (BMI ≥ 35) [26,27]. One study observed no difference in in-hospital deaths between normal and overweight (BMI > 28) patients, but overweight patients did show more severe disease symptoms [28]. Because BMI does not discriminate between fat and lean body mass and poorly reflects fat distribution, four studies used measures of visceral adipose tissue (VAT) obtained by high-resolution computed tomography [42,43,44,45]. In COVID-19 patients reporting at the emergency department, subsequent admission to the ICU was associated with 30% higher VAT and 30% lower subcutaneous adipose tissue (SAT) [43] with VAT scores being the best ICU admission predictor. Another study observed no differences in BMI but did see higher VAT and a higher VAT/total adipose tissue (TAT) ratio in hospitalized patients compared to outpatients that did not require hospitalization [42]. Finally, in a different study it was observed that both VAT and TAT were associated with ICU admission [45]. To summarize, numerous studies indicate that obesity is associated with increased disease severity and mortality in COVID-19.

## 3. Underlying Mechanisms Linking Obesity to Major Complications as a Result of SARS-CoV-2 Infection

To gain a better understanding of the pathophysiology of COVID-19 in patients suffering from obesity, in the next section and in Figure 1 we provide an overview of the mechanistic pathways linking obesity with COVID-19 disease severity, with a focus on the metabolic and immunological consequences of obesity on COVID-19 disease course. Other features of obesity, like impaired respiratory mechanics and pulmonary function and the co-existence of metabolic disorders like diabetes and cardiovascular disease within a single individual also increase the risk for severe COVID-19 and complications [46], but are beyond the scope of this review. In this overview we discuss, *in vitro*, animal and human *in vivo* studies, including clinical trials. Studies were excluded when not indexed and when methodology did not reach minimal criteria.

### 3.1. Obesity Facilitates Viral Entry of SARS-CoV-2

Cell entry of SARS-CoV-2 virus depends on binding of the viral spike (S) proteins to cellular receptors and on S protein priming by host cell proteases. Obesity can promote viral entry by the upregulation of some of these host cell receptors.

#### 3.1.1. Obesity Increases ACE2 Expression

Viral entry into target cells is facilitated by binding of the SARS-CoV-2 spike (S) protein to the membrane protein angiotensin-converting enzyme 2 (ACE2) and the priming of the S protein by serine protease TMPRSS2 and the endosomal cysteine proteases cathepsin B and L expressed by host cells [47,48]. ACE2 is part of the renin-angiotensin system (RAS) and plays an important role in regulating systemic blood pressure. Renin induces proteolytic cleavage of angiotensinogen into angiotensin (Ang)-I, which is then converted into Ang-II by ACE1. The action of ACE1 induces vasoconstriction and inflammation while ACE2 acts as a counter-regulator of the RAS. ACE2 induces vasodilatation and has an anti-inflammatory effect, by converting the pro-inflammatory Ang-II into Ang-1-7 which opposes the actions of Ang-II. ACE2 is expressed in a wide variety of human tissues such as the small intestine, testis, kidneys, heart, thyroid, lungs and brain [49]. Adipose tissue also expresses ACE2, together with all other components of the RAS. The RAS has been shown to play an important role in lipid metabolism and vice versa [50,51,52,53]. For example, the expression of ACE2 in adipocytes seems to be promoted by a high-fat diet [54]. Also, analysis of available transcriptome data showed that diet-induced obese mice displayed increased expression of ACE2 in the lung epithelium [55]. The expression of ACE2 was correlated with the expression of genes that code for sterol response element binding proteins 1 and 2, transcription factors controlling lipid synthesis and adipogenesis [55].

In addition, a number of studies indicate that ACE2 expression can be upregulated by a wide range of pro-inflammatory cytokines [56,57]. The chronic inflammatory state that characterizes obesity could therefore potentially facilitate viral entry [56]. Moreover, it has been suggested that viral pools can lodge in adipose tissue and promote shedding, immune activation, and chronic excessive cytokine release [58]. Earlier studies demonstrated that other viruses such as influenza A, HIV and cytomegalovirus use the adipose tissue as a reservoir [59].

Binding of SARS-CoV-2 to ACE2, which prevents ACE2 from exerting its enzymatic activities [60], has further consequences for immune system functioning. Decreased enzymatic ACE2 activity results in a reduction of anti-inflammatory Ang-1-7 and an accumulation of pro-inflammatory Ang-II. This process further increases the risk for COVID-19-related immune system complications such as ARDS.

In conclusion, obesity and its related pro-inflammatory state can promote viral entry, viral shedding and excessive immune activation through the upregulation of ACE2.

#### 3.1.2. Activation of the Unfolded Protein Response by GRP78

Another receptor that provides viral entry for SARS-CoV-2 is glucose-regulated protein 78 (GRP78) [61,62]. GRP78, also called immunoglobulin heavy chain binding protein (BiP), is part of the heat shock protein 70 family and is a molecular chaperone found in the lumen of the endoplasmic reticulum (ER). GRP78 functions as an anti-apoptotic regulator that protects cells against cell death induced by ER stress [62]. GRP78 protects against cell death by ensuring the correct folding and assembly of proteins in the ER and by initiating the degradation of misfolded proteins. This evolutionarily preserved cellular homeostatic response to ER stress, also named the ER-stress or unfolded protein response (UPR) [63] can be disrupted by viruses in the establishment of acute, chronic and latent infections [64].

Normally, new foreign viral proteins produced inside host cells would be immediately degraded by the UPR. Therefore, viruses that acquired the ability to disrupt the process designed to degrade unrecognized protein have a clear advantage. Binding GRP78 is a mechanism commonly used by viruses like the Ebola virus, Zika virus, Dengue virus, Japanese encephalitis virus, Coxsackie A9, MERS-CoV and the Borna disease virus to ensure safe entry into host cells and to facilitate viral replication [65,66,67,68,69,70,71]. Also, the S proteins of the SARS-CoV-2 virus binds GRP78 with high affinity [61,72].

Obesity and various factors associated with obesity induce ER stress [73] and consequently UPR activation. These factors, including hypoxia, reactive oxygen species, insulin resistance, nutritional imbalance and excessive fat storage [74,75,76], stimulate GRP78 expression as a mechanism of the UPR to restore normal cell functions [77,78]. For example, evidence from *in vitro* and *in vivo* studies shows that dyslipidemia is associated with GRP78 overexpression in adipocytes (especially in white adipose tissue) [79], pneumocytes [80], neurons of the hypothalamus [81] and hepatocytes [82]. 

GRP78 overexpression induces its translocation to the cell membrane or cell surface. When GRP78 is located on the cell surface, it is referred to as cell surface (cs)-GRP78. This acts as a multi-functional receptor that can bind various ligands and plays a crucial role in apoptosis and inflammation, among other things [62]. By binding to cs-GRP78, SARS-CoV-2 could ensure safe entry into host cells [61,72].

#### 3.1.3. Heparan Sulfate Proteoglycans and Neuropilin-1

Heparan sulfate proteoglycans (HSPGs) are located in the extracellular matrix and at the surface of the cell, where they act as co-receptors for various ligands. HSPGs are widely expressed and mediate many biological activities, including angiogenesis, blood coagulation, developmental processes, and cell homeostasis. The binding of cytokines, chemokines and growth factors to HSPGs at the cell surface prevents their degradation [83]. A significant amount of research has established that many different types of viruses can interact with HSPGs. For some of these viruses, these interactions are essential for internalization into host cells [84]. This is also the case for the internalization of SARS-CoV-2 [85].

Syndecans are the major family of transmembrane HSPGs and are present on virtually all nucleated human cells. Syndecans have been shown to facilitate the cellular entry of SARS-CoV-2 *in vitro*. Among syndecans, syndecan-4 was most efficient in mediating SARS-CoV-2 uptake, yet overexpression of other isoforms, including syndecan-1 and the neuronal syndecan-3, also increased SARS-CoV-2 internalization [85,86]. Syndecan 4 is widely expressed in most adult tissues [87] and, among other functions, is involved in lipid metabolism, by clearing pro-atherogenic remnant lipoproteins from the circulation [88]. In addition, syndecan-4 is crucial for adipocyte differentiation and proliferation [89]. Syndecan 4 is upregulated upon activation of the pro-inflammatory transcription factor nuclear factor-κB (NF-κB) [90,91] and its expression is therefore markedly increased under inflammatory conditions [92]. 

Neuropilin-1 (NRP-1), a membrane-bound co-receptor for growth factors such as vascular endothelial growth factor (VEGF), also facilitates viral entry of SARS-CoV-2 *in vitro* [93]. NRP-1 is strongly involved in adipogenesis and is highly upregulated during adipose-derived stem cell differentiation [94]. 

In obesity, chronic inflammation, higher circulating pro-atherogenic lipoprotein levels [95] and the constant expansion of adipose tissue accompanied with the differentiation and proliferation of adipose-derived stem cells, could increase syndecan (especially syndecan-4) and NRP-1 expression in various tissues. The up-regulation of syndecans and NRP-1 could facilitate cellular entry of SARS-CoV-2. However, further studies need to confirm this theory.

### 3.2. Obesity Related Insulin Resistance Contributes to an Impaired Immune Response to SARS-CoV-2 Infection

#### 3.2.1. Obesity Induces Insulin Resistance 

Insulin resistance is a consequence of the impairment of insulin signaling in insulin-responsive cells, like hepatocytes, myocytes and adipocytes. In the adipose tissue, insulin resistance is promoted by excess lipid accumulation, causing hypoxia and inflammation. The subsequent invasion of macrophages in the adipose tissue that release pro-inflammatory cytokines further impair insulin signaling (see also Section 3.2.2). As a consequence of adipocyte insulin resistance higher levels of free fatty acids (FFA) leave the fat tissue and enter into the circulation. The increase in circulating FFA and pro-inflammatory mediators further impairs insulin action in other metabolically active organs and tissues, including skeletal muscle and the liver, leading to systemic insulin resistance, which is associated with impaired glucose transport [96,97]. It has been shown that the size of the visceral adipose tissue and adipocyte size in humans is directly associated with systemic insulin resistance [14]. 

#### 3.2.2. Insulin Resistance Is Induced by Adipocytes and Related Immune Cells 

Adipose tissue macrophages can be divided into the classical M1 and alternatively activated M2 macrophages. Adipose tissue from obese individuals contains elevated numbers of M1-like macrophages, which produce pro-inflammatory cytokines, such as TNF-α and IL-6 [98,99]. These pro-inflammatory cytokines inhibit insulin signaling pathways in multiple tissues [100]. 

The production of TNF-α by M1 macrophages is positively related to the size of the adipose tissue mass. In the adipose tissue TNF-α induces phosphorylation and inactivation of insulin receptors and activates lipolysis, which increases the FFA load. The production of IL-6 by adipocytes and related immune cells is also associated with the amount of body fat. IL-6 induces the production of the pro-inflammatory acute-phase protein C-reactive protein (CRP) and increases fibrinogen levels, resulting in a prothrombotic state. It also promotes adhesion molecule expression by endothelial cells and activates local RAS pathways [101].

Chronic elevations of pro-inflammatory cytokines, such as TNF-α and IL-6, also directly influence COVID-19 disease course. Because TNF-α plays an important role in promoting ARDS, obese individuals with chronically elevated serum TNF-α are at greater risk of developing this life-threatening complication [102]. In patients with COVID-19, IL-6 levels are significantly elevated and associated with adverse clinical outcomes. Meta-analysis of mean IL-6 concentrations demonstrated 2.9-fold higher levels of IL-6 in hospitalized COVID-19 patients with complications compared to patients without complications [103]. 

#### 3.2.3. Systemic Insulin Resistance Impairs the Immune Response 

Insulin acts upon immune cells and therefore, systemic insulin resistance can have a substantial impact on the functioning of the adaptive and innate immune system [100]. Animal studies have shown that insulin signaling is essential for optimal T cell effector function [104]. In humans, insulin-resistant individuals displayed delayed innate immune-related pathway activation after respiratory viral infection compared to insulin-sensitive individuals, suggesting an impairment of the early acting innate immune response [105]. 

It is likely to assume that chronic inflammation and impairments of the immune response as a consequence of obesity-induced insulin resistance may reduce efficient viral clearance and drive organ injury in the development of severe COVID-19 [106]. 

### 3.3. Disrupted Leptin Signaling in Obesity can Induce Hyperinflammation during SARS-CoV-2 Infection

#### 3.3.1. Leptin Is an Important Regulator of Energy Metabolism

Leptin is a pleiotropic protein secreted primarily from white adipose tissue into the bloodstream and can be transported across the blood-brain barrier. Through its effects on the central nervous system (CNS) and peripheral tissues, leptin is an important regulator of energy homeostasis, metabolism, neuroendocrine and immune system function [107]. 

Obesity proceeds from a chronic energy imbalance and is characterized by persistent hyperleptinemia and central leptin resistance. Under physiological conditions, leptin informs the brain about the energy status in the periphery, but in obesity, signaling to regulatory centers in the brain that normally inhibit food intake and regulate body weight and energy homeostasis is disrupted. Mechanisms underlying leptin resistance, include disruption of leptin signaling in hypothalamic and other CNS neurons, impaired leptin transport across blood-brain barrier, hypothalamic inflammation, ER stress, and autophagy [107,108].

#### 3.3.2. Leptin Modulates the Immune System

Leptin has pro-inflammatory properties and upregulates the secretion of pro-inflammatory cytokines [18,22]. Leptin signaling results in the modulation of both the innate and adaptive immune system on multiple levels. Leptin signaling occurs primarily through the binding of leptin to the long isoform of the leptin receptor, followed by activation of the JAK/STAT pathway [2]. In the innate immune system, increased leptin production stimulates chemotaxis and neutrophil survival, induces pro-inflammatory cytokine production [109,110], and higher expression of adhesion molecules by eosinophils and basophils [111,112]. Monocyte activation and proliferation as well as the production of pro-inflammatory cytokines and chemotaxis are also stimulated by leptin [22]. Certain immune cells, more specifically those that contain the long isoform of the leptin receptor, may become unresponsive to leptin, or leptin resistant, when exposed to high leptin levels for an extended period of time [113,114]. Therefore, chronic hyperleptinemia, as seen in obesity, can have detrimental effects on the immune response due to both chronic pro-inflammatory effects and immune cell dysfunction [21]. 

Additionally, the adaptive immune system responds to leptin. Leptin induces a shift towards the more pro-inflammatory Th1 response [21], activates T and B lymphocytes, and inhibits regulatory T cells [18,115,116]. Regulatory T cells are involved in suppressing an excessive immune response [117,118]. It was demonstrated in patients suffering from obesity that leptin levels and BMI were inversely correlated with the number of regulatory T cells [119].

In obesity, leptin levels can be further increased due to infection or sepsis [120]. Elevated leptin levels, in combination with the obesity-induced pro-inflammatory state, further increases the risk for the development of a disproportionate or hyper-inflammatory response upon SARS-CoV-2 infection. Furthermore, SARS-CoV-2 infection has been shown to increase the expression of the gene that encodes for suppressor of cytokine signaling 3 (SOCS3) in lung epithelium [55]. This gene is a key regulator of inflammation and an inhibitor of leptin signaling. Increased SOCS3 expression as a result of SARS-CoV-2 infection could therefore further impair leptin signaling and negatively influence the immune response in patients suffering from obesity [121].

### 3.4. Hyperferritinemia as a Result of Hyperinflammation can Induce a Cytokine Storm

Ferritin is an iron-binding molecule that stores iron in a biologically available form for vital cellular processes and protects proteins, lipids and DNA from the potential toxicity of this metal element. Ferritin is a marker of the acute-phase response, and its secretion is regulated by pro-inflammatory cytokines. However, the origin of circulating serum ferritin during inflammatory conditions is still debated [122]. While some describe serum ferritin as a leakage product of damaged cells [123], increasing evidence shows that circulating serum ferritin levels may play a critical role in the inflammatory process [91]. Serum ferritin may protect the host during active infection by limiting the availability of iron to pathogens, a phenomenon called ‘nutritional immunity’ [124,125,126]. 

COVID-19 is also accompanied by a rise in circulating ferritin levels. Serum ferritin levels can be used as a diagnostic marker and even a predictor of COVID-19 severity [127,128,129,130,131]. The elevated ferritin levels observed in COVID-19 patients are probably a consequence of the inflammatory process induced by SARS-CoV-2 infection and actively act as enhancer of the inflammatory process in more severe COVID-19 [132]. Alternatively, it has been suggested that ferritin levels increase due to the break-down of red blood cells by the SARS-CoV-2 virus. By breaking down red blood cells and then attacking the hemoglobin 1-beta chain, the virus could separate iron from the porphyrin ring to eventually hijack the porphyrin ring, producing a rise in free iron and subsequently increasing ferritin levels [133]. Although hemoglobin-related biomarkers such as serum ferritin, progressively increase as the severity of COVID-19 increases [134,135,136], there is very limited evidence supporting this hypothesis [133]. This hypothesis has subsequently been refuted by DeMartino and colleagues, confirming that disease markers such as hemoglobin, iron and ferritin did not differ between critically ill COVID-19 patients and non-COVID ARDS patients [137]. 

Several studies have shown a relationship between obesity and elevated ferritin serum levels [138]. Chronic inflammation, as a result of the increased release of leptin and pro-inflammatory cytokines by adipocytes and related immune cells, [11,21,23,120] likely contributes to the manifestation of elevated ferritin levels in individuals with obesity.

Infection with SARS-CoV-2 in individuals with obesity as pre-existing condition can progress into a hyperferritinaemic state as both SARS-CoV-2 and obesity initiate the release of ferritin. Ferritin further stimulates macrophages to produce pro-inflammatory cytokines, mainly IL-1, IL-6 and IL-17. At overly elevated serum ferritin levels, macrophages produce so many cytokines that the situation can evolve into a cytokine storm [122]. The cytokine storm occurring with COVID-19 could be considered a hyperferritinemia syndrome [139] and is a significant adverse development in the course of the disease, translating into a marked increased risk of death [140]. 

### 3.5. Obesity-Related Risk Factors for Developing Coagulopathy in COVID-19 Patients

About a third of ICU patients with COVID-19 develop thrombotic complications [141,142]. The characteristics of CAC seem to be more complex than the development of thromboinflammation triggered by systemic inflammation in response to infection [143]. For example, the presence of systemic microthrombi and hemorrhage in SARS-CoV-2 affected organs indicates a malfunction in the coordination of coagulation and fibrinolysis [144]. Moreover, CAC is commonly associated with increased D-dimer and fibrinogen levels, indicating there is initially no suppression of fibrinolysis [145]. Abnormalities in other coagulation biomarker such as prothrombin time and platelet count are less frequent and seem less affected by SARS-CoV-2 infection [146].

In the following section we provide an overview of parameters that are dysregulated in obesity and could contribute to increase the risk of developing CAC (See also Figure 1).

#### 3.5.1. Hyperleptinemia

Hyperleptinemia is a risk factor for developing thrombi. Leptin affects blood clotting by enhancing prothrombotic and antifibrinolytic protein expression in vascular and inflammatory cells and thereby producing a more hypofibrinolytic state [147]. A large population-based cohort study from The Netherlands, examining associations between serum leptin concentrations and coagulation factor concentrations and parameters of platelet activation, showed that serum leptin concentrations were positively associated with concentrations of coagulation factor VIII and IX [148]. Hyperleptinemia and leptin resistance have also been described as risk factors for the development of cardiovascular diseases [149,150]. As elevated leptin levels are a key feature of obesity [149,151,152], the prothrombotic state produced by hyperleptinemia, increases the risk for developing thrombotic complications in COVID-19.

#### 3.5.2. PAI-1 Production by Adipocytes

Plasminogen activator inhibitor 1 (PAI-1) is the primary physiological inhibitor of plasminogen activation. Elevations in plasma PAI-1 compromise normal fibrin clearance mechanisms and promote thrombosis. PAI-1 is also produced by adipocytes and its production is dramatically upregulated in obesity [153,154]. PAI-1 mRNA expression was demonstrated in the visceral and subcutaneous fat of obese rats [155] and in adipose tissue from human subjects [156]. In both rats and humans, VAT produced significantly more PAI-1 compared to SAT. These results are consistent with the observation that cardiovascular risk is most closely correlated with central obesity [157]. PAI-1 produced by adipocytes and vascular endothelium is involved in tissue expansion and angiogenesis necessary during adipose tissue development [158]. 

Several molecular mechanisms are involved in the upregulation of PAI-1 mRNA expression in obesity. For example, different cytokines including TNF-α and transforming growth factor (TGF)-ß, triglycerides, free fatty acids and insulin all stimulate PAI-1 expression in adipose tissue [153]. Leptin, additionally, contributes to a prothrombotic state by increasing the expression of PAI-1 in vascular endothelium [159]. Thus, obesity-induced upregulation of PAI-1 further increases the risk for coagulopathy, following SARS-CoV-2 infection.

#### 3.5.3. Endothelial Dysfunction

The endothelium serves as a dynamic barrier that separates blood from interstitia. Endothelial cells respond rapidly to changes in the circulation and become activated according to environmental needs [160]. The multitude of physiological functions of the endothelium are still a topic of extended research. Endothelium is recently described as an active regulator of lipid and glucose homeostasis [161]. 

Endothelial cells play a major role in vascular homeostasis and blood coagulation. Endothelial dysfunction is considered a hallmark of metabolic diseases and is characterized by a loss of molecular cell functions and inevitably causing coagulopathy. Endothelial dysfunction contributes to various pathological states such as atherothrombosis, arterial thrombosis (stroke, visceral and peripheral artery occlusive diseases), venous thrombosis, intravascular coagulation and thrombotic microangiopathies [160]. 

Oxidative stress is considered the major cause of endothelial dysfunction. Obesity generates oxidative stress through different pathways, such as hyperglycemia, known to trigger vascular damage by inducing the accumulation of reactive oxygen species (ROS) [162]. Hyperglycemia also activates NF-κB, a transcription factor that mediates vascular inflammation [162]. The exacerbated production of pro-inflammatory cytokines by adipose tissue further increases oxidative stress levels and promotes the upregulation of procoagulant factors and adhesion molecules in the endothelium, the downregulation of anticoagulant regulatory proteins, increases thrombin generation, and enhances platelet activation [163]. Not coincidentally, endothelial dysfunction is a common feature of comorbidities that increase the risk for severe COVID-19, including hypertension, obesity, diabetes mellitus, coronary artery disease and heart failure. 

Furthermore, because ACE2 is expressed abundantly on vascular endothelial cells of both small and large arteries and veins [164], SARS-CoV-2 infection of endothelial cells can further aggravate endothelial dysfunction. Endothelial damage and dysfunction may thus be the result of cellular infection by SARS-CoV-2, as well as a consequence of obesity-associated excessive systemic inflammation [165]. Evidently, patients with preexisting endothelial dysfunction are more vulnerable to develop severe complications, including coagulopathy, following SARS-CoV-2 infection (See also Figure 1).

#### 3.5.4. Vitamin K

Vitamin K is a fat-soluble vitamin, required for the carboxylation of vitamin K-dependent proteins (VKDP). Intrahepatic VKDP include coagulation factors II, VII, IX, X, and different anticoagulant proteins. Extra-hepatic VKDP include diverse gamma-carboxyglutamate (Gla) proteins involved in maintaining bone homeostasis, as well as inhibiting ectopic calcification [166]. Vitamin K is not a single entity but a family of structurally related molecules such as phylloquinone, also referred to as vitamin K1, and menaquinone or vitamin K2.

Obesity is linked with vitamin K deficiency [167,168,169]. Vitamin K deficiency results in decreased vitamin K-dependent carboxylation and phosphorylation of Gla-proteins and, as a consequence, into elevated levels of circulating desphosphorylated-uncarboxylated matrix Gla protein (dp-ucMGP). Recent research studied the relationship between obesity and serum dp-ucMGP in a cohort of 278 Chinese Han people. The results demonstrated that serum dp-ucMGP level was positively associated with visceral fat index, waist height ratio, but not BMI [167]. In addition, lower vitamin K2 levels were observed in obese hemodialysis patients compared to non-obese patients [168]. Also, a lower vitamin K1 status was observed in patients with obesity compared to healthy individuals [169]. Yet, differences in dietary K1 intake could not fully explain this observation. It has been suggested that vitamin K accumulates in adipose tissue, thereby reducing the bioavailability of this vitamin in individuals with obesity [169]. This theory could be plausible since a similar mechanism has been established for other fat-soluble vitamins [170,171]. However, understanding the mechanisms underlying the association between obesity and vitamin K deficiency remains a topic for further investigation.

Vitamin K status was also assessed in COVID-19 patients by measuring dp-ucMGP. Levels of dp-ucMGP were significantly elevated in COVID-19 patients compared to controls and were higher in COVID-19 patients with unfavorable disease outcome. Carboxylated matrix Gla-protein protect against degradation and calcification of vasculature and elastic fibers in the extracellular matrix of the lungs. In COVID-19, elastic fiber degradation combined with calcification of these fibers due to low vitamin K status could aggravate lung injury and lung fibrosis [172]. 

Thus, to summarize, we can argue that several factors associated with obesity increase the risk for CAC, including elevated leptin levels, increased PAI-1 levels, endothelial dysfunction and low vitamin K levels. The latter is possibly also involved in the development of more severe lung injury in COVID-19. 

## 4. Conclusions and Implications for Further Research

Obesity cannot simply be defined as an excess of fat cells. Adipose tissue releases many active substances, such as adipokines and components of the RAS, all influencing the brain and metabolic- and immune system. Being obese increases the risk of SARS-CoV-2 infection and complications via several mechanisms. First, viral entry is enhanced due to increased ACE2, csGRP78 and presumably HSPG and NRP-1 expression levels in various cell types, like pneumocytes and adipocytes. Second, the immune system is unable to provide an adequate immune response leading to impaired viral clearance. Eventually, the immune system can overreact as a result of pro-inflammatory ‘priming’ due to excessive cytokine production by adipose tissue and its related immune cells and high ferritin levels, eventually triggering a cytokine storm [122,139]. Finally, hyperleptinemia, PAI-1 production by adipocytes and endothelial cells, endothelial dysfunction and low levels, or low bioavailability, of vitamin K, all increase the risk for the development of thrombus formation and hemorrhage. 

An important lesson learned from the coronavirus pandemic is the importance of a healthy lifestyle to positively influence the course of COVID-19 disease. A non-processed nutrient-rich diet [173,174,175], limited excessive or overly energy-rich food, sufficient and intensive exercise [176], sufficient sleep [177] and avoiding chronic psycho-emotional stress [178] are all efficient health-promoting measures in the prevention of obesity [179]. We also advocate an integrated multidisciplinary approach in the fight against COVID-19. Future research must identify causes of severity and complications to develop efficient preventive measures and curative interventions.

## Figures and Tables

**Figure 1 cells-10-00933-f001:**
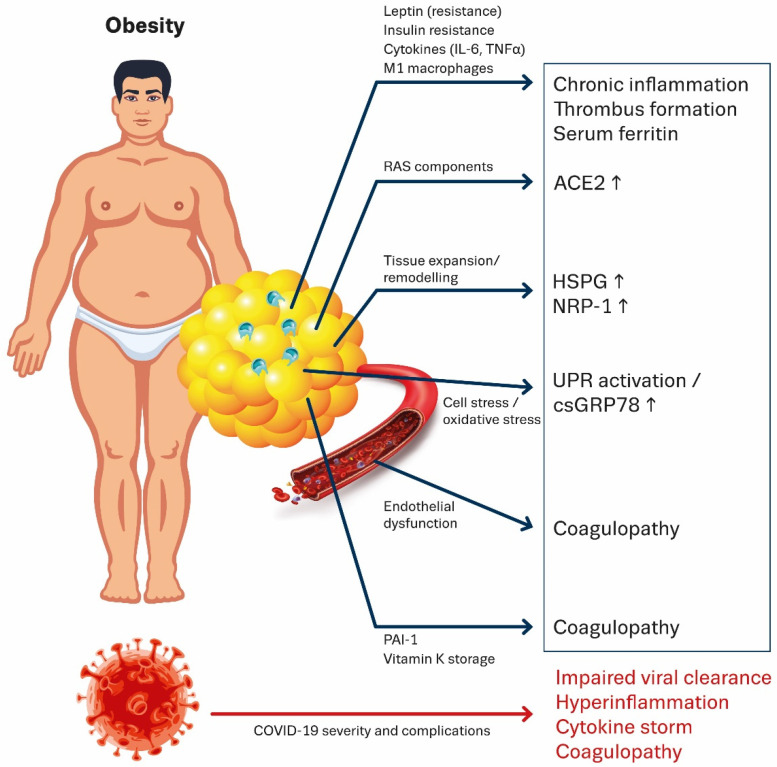
Schematic representation of different mechanisms through which obesity can promote COVID-19 disease severity and risk for complications. Obesity is often accompanied by insulin and leptin resistance which impairs viral clearance. Next, obesity is characterized by large hypoxic adipocytes infiltrated with immune cells and M1 macrophages leading to a chronic inflammatory state, hypercoagulability and hyperferritinemia. ACE2 produced by adipocytes could provide viral entry into the adipose tissue. In this way the adipose tissue could possibly function as a reservoir for the virus. The constant tissue expansion and tissue remodeling accompanying obesity in concert with high cell stress can upregulate the expression of other potential SARS-CoV-2 receptors, such as csGRP78, HSPG and NRP-1 in adipose tissue and other organs. Obesity-associated endothelial dysfunction, enhanced production of PAI-1 and vitamin K deficiency all increase the risk of developing COVID-19 associated coagulopathy. Abbreviations: IL-6 = Interleukin-6; TNFα = Tumor necrosis factor α; RAS = Renin angiotensin system; ACE = Angiotensin converting enzyme; HSPG = Heparan sulfate proteoglycan; NRP-1 = Neuropilin-1; UPR = Unfolded protein response; csGRP78 = Cell surface glucose related protein 78; PAI-1 = Plasminogen activator inhibitor 1.

**Table 1 cells-10-00933-t001:** Clinical Studies Investigating the Association Between Obesity and Disease Severity in COVID-19 Patients.

Reference	Study Design	Predictor	Outcome	Effect
Anderson et al., 2020 [29]	Retrospective cohort study (*n* = 2466)	BMI categories: underweight (BMI < 18.5), normal weight (BMI ≥ 18.5 to <25), overweight (BMI ≥ 25 to <30), class 1 obesity (BMI ≥ 30 to <35), class 2 obesity (BMI ≥ 35 to <40), and class 3 obesity (≥ 40)	Intubation death	Patients younger than 65 with obesity were at higher risk for intubation or death, with the highest risk among those with class 3 obesity (BMI ≥ 40).
Battisti et al., 2020 [43]	Cohort study (*n* = 441)	VAT/SAT ratio	ICU admission	VAT/SAT was associated with increased risk of ICU admission.
Chandarana et al., 2020 [42]	Retrospective study (*n* = 51)	VAT, SAT, TAT, VAT/TAT and BMI	Hospi- talization	Higher VAT and VAT/TAT in hospitalized patients.
Deng et al., 2020 [30]	Retrospective cohort study (*n* = 65)	BMI, subcutaneous fat thickness, epicardial fat and visceral fat	Disease severity	High BMI was a risk factor for severe COVID-19.
Frank et al., 2020 [25]	Retrospective cohort study (*n* = 305)	BMI categories: BMI < 25, BMI ≥ 25 to < 30, BMI ≥ 30 to < 35, and BMI ≥ 35	Intubation, death	BMI ≥ 30 was associated with an increased risk of intubation or death.
Hernàndez- Galdamez et al., 2020 [38]	Cross-sectional study (*n* = 212,802)	Obesity (not specified)	Hospi- talization ICU admission Intubation death	Obesity was associated with an increased risk of hospitalization, ICU admission, intubation and death.
Kalligeros et al., 2020 [31]	Retrospective cohort study (*n* = 103)	BMI categories: BMI < 25, BMI ≥ 25 to <30, BMI ≥ 30 to <35, BMI ≥ 35	ICU admission, IMV	Severe obesity (BMI ≥ 35) was positively associated with ICU admission. Obesity (BMI ≥ 30 to <35) and severe obesity (BMI ≥ 35) were positively associated with the use of IMV.
Kim et al., 2020 [26]	Retrospective cohort study (*n* = 10,861)	BMI categories: underweight (BMI < 18.5), normal weight (BMI ≥ 18.5 to <25), overweight (BMI ≥ 25 to < 30), obesity class Ⅰ (BMI ≥ 30 to < 35), obesity class Ⅱ (BMI ≥ 35 to <40), and obesity class Ⅲ (BMI ≥ 40)	IMV, death	Categories: overweight, obesity class Ⅰ, Ⅱ and Ⅲ were associated with increased risk of requiring IMV. Underweight and obesity classes Ⅱ and Ⅲ were associated with increased risk of death.
Mash et al., 2021 [32]	Descriptive observational cross-sectional study (*n* = 1376)	BMI categories: normal (BMI ≥ 18.5 to <25), overweight/obese (BMI ≥ 25)	Death	Overweight/obesity (BMI ≥ 25) was significantly linked with mortality.
Nakeshbandi et al., 2020 [24]	Retrospective cohort study (*n* = 504)	BMI categories: normal (BMI ≥ 18.5 to <25), overweight (BMI ≥ 25 to <30), and obese (BMI ≥ 30)	Mortality, intubation	Patients with overweight and obesity were at increased risk for mortality and intubation.
Palaiodimos et al., 2020 [27]	Retrospective cohort study (*n* = 200)	BMI categories: BMI < 25, BMI ≥ 25 to <35, BMI ≥ 35	In-hospital mortality, Worse in-hospital outcomes	Severe obesity (BMI ≥ 35) was associated with higher in-hospital mortality and worse in-hospital outcomes.
Parra- Bracamonte et al., 2020 [39]	Cross-sectional study (*n* = 331,298)	Obesity (not specified)	Death	Obesity was associated with higher risk of mortality.
Pediconi et al., 2020 [44]	Retrospective cohort study (*n* = 62)	VAT score (overweight: VAT area 100–129 cm^2^ or VAT score 1. Obesity: VAT area ≥ 130 cm^2^ or VAT score 2)	ICU admission	VAT score was the best ICU admission predictor.
Peña et al., 2020 [40]	Cross-sectional study (*n* = 323,671)	Obesity (not specified)	Death	Obesity was a major risk factor for mortality.
Randhawa et al., 2021 [33]	Retrospective cohort study (*n* = 302)	BMI categories: normal weight (BMI < 30), obesity BMI ≥ 30)	Compli- cations	Patients with obesity were more likely to suffer severe complications.
Rao et al., 2020 [28]	Retrospective cohort study (*n* = 240)	BMI (overweight, BMI > 28)	In-hospital death, Disease severity	Being overweight was related to COVID-19 severity but not to in-hospital death.
Salinas Aguirre et al., 2021 [41]	Cross-sectional study (*n* = 17,479)	Obesity (not specified)	Death	Obesity was associated with mortality.
Simonnet et al., 2020 [34]	Retrospective cohort study (*n* = 124)	BMI categories: lean (BMI ≥ 18.5 to <25), overweight (BMI ≥ 25 to < 30), moderate obesity (BMI ≥ 30 to < 35) and severe obesity (BMI ≥ 35)	Need for IMV	Need for IMV was associated with BMI.
Suleyman et al., 2020 [35]	Case series (*n* = 463)	BMI categories: severe obesity (BMI ≥ 40)	ICU admission	Severe obesity was independently associated with ICU admission.
van Zelst et al., 2020 [37]	Prospective observational cohort study (*n* = 166)	BMI Abdominal adiposity (waist-to-hip-ratio)	Unfavorable outcome	Abdominal adiposity and BMI were associated with an increased risk for unfavorable outcome (respiratory support of 3 L/min, intubation, ICU admission).
Watanabe et al., 2020 [45]	Retrospective cohort study (*n* = 150)	TAT VAT	ICU admission	TAT and VAT had a univariate association with ICU admission.
Zhu et al., 2020 [36]	Retrospective cohort study (*n* = 489,769)	BMI, categories: normal weight (BMI ≥ 18.5 to <25), overweight (BMI ≥ 25.0 to <30), and obese (BMI ≥ 30); waist circumference and waist-to-hip ratio	Hospi- talization with ‘severe COVID-19′	BMI, waist circumference and waist-to-hip ratio were positively associated with the risk of severe COVID-19.

Abbreviations: VAT = visceral adipose tissue; SAT = subcutaneous adipose tissue; TAT = total adipose tissue; BMI = body mass index (kg/m^2^); IMV = invasive mechanical ventilation; ICU = intensive care unit.

## Data Availability

Not applicable.

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
