# Peer review of "Obesity as a Risk Factor for Severe COVID-19 and Complications: A Review"

_cells, 2021, doi:10.3390/cells10040933_

Round 1

Reviewer 1 Report

The manuscript submitted by Demeulemeester F. et al. "Obesity as a risk factor for severe COVID-19 and complications" elegantly discusses the possible implications and the main mechanisms leading to increased COVID-19 severity and its complications in obese individuals. 

I only have a few minor comments/suggestions.

  • In the section 3, I would recommend to include a subsection on the role of insulin resistance/insulin singling and another subsection about specific cytokines, such IL-6, and TNF alpha, since those are crucial for insulin resistance in obese individuals, are maintained by the chronic low grade inflammation in obesity, and play a role on cytokine storm observed during severe COVID-19. And also because IL-6 is one of the targets for treatment.
  • Another section regarding on glucose and lipid metabolism since those are also altered during obesity would increase even more the quality of the discussion. 

Author Response

Cells-1167254

Demeulemeester et al., “Obesity as a risk factor for severe COVID-19 and complications: a review”

Responses to reviewers’ comments.

We appreciate the thoughtful reviews by the referees. They have been very helpful in improving the quality of our manuscript. We have addressed each of the specific comments below and in the revised manuscript. 

Reviewer 1

The manuscript submitted by Demeulemeester F. et al. "Obesity as a risk factor for severe COVID-19 and complications" elegantly discusses the possible implications and the main mechanisms leading to increased COVID-19 severity and its complications in obese individuals. I only have a few minor comments/suggestions.

Comment #1:

In the section 3, I would recommend to include a subsection on the role of insulin resistance/insulin singling and another subsection about specific cytokines, such IL-6, and TNF alpha, since those are crucial for insulin resistance in obese individuals, are maintained by the chronic low grade inflammation in obesity, and play a role on cytokine storm observed during severe COVID-19. And also because IL-6 is one of the targets for treatment.

Response: Thank you for pointing this out. We now included subsection 3.2.1 where we discuss how obesity induces impaired insulin signaling/insulin resistance. In subsection 3.2.2 we explain how the pro-inflammatory cytokines TNF-α and IL-6 contribute to insulin resistance and play a role in the development of severe COVID-19. In section 3.2.1 we further discuss the consequences of insulin resistance and associated chronic inflammation on the functioning of the immune system.

Comment #2:

Another section regarding on glucose and lipid metabolism since those are also altered during obesity would increase even more the quality of the discussion

Response: In section 3.2.1 we now discuss the consequences of impaired lipid metabolism on the development of insulin resistance and impaired glucose transport (line 237-242): ''As a consequence of adipocyte insulin resistance higher levels of free fatty acids (FFA) leave the fat tissue and enter into the circulation. The increase in circulating FFA and pro-inflammatory mediators further impairs insulin action in other metabolically active organs and tissues, including skeletal muscle and the liver, leading to systemic insulin resistance, which is associated with impaired glucose transport."

Reviewer 2 Report

In this review Demeulemeester et al. summarize clinical findings on obesity and COVID-19 disease outcomes, discussing also the plausible mechanisms underlying the link between obesity and major disease complications as a result of SARS-CoV-2 infection.

Some minor issues need to be addressed:

  1. in the title and in the aim (line 36-40) should be clarify that author performed a review.
  2. Search strategy should be included.
  3. Obesity has detrimental effects on respiratory mechanics, furthermore it may increase the risk of many metabolic and chronic disease (diabetes, cardiovascular disease, cancer...) that may co-exist in a single individual. These aspects may further influence mechanisms involved in covid-19 disease severity. Please add this aspect in the paper.

Author Response

Reviewer 2:

In this review Demeulemeester et al. summarize clinical findings on obesity and COVID-19 disease outcomes, discussing also the plausible mechanisms underlying the link between obesity and major disease complications as a result of SARS-CoV-2 infection. Some minor issues need to be addressed:

Comment #1:

in the title and in the aim (line 36-40) should be clarify that author performed a review.

Response: We now changed the title in: “Obesity as a risk factor for severe COVID-19 and complications: a review”. Also in the aim (line 40) we made clear we performed a review:  “In order to facilitate a better insight into the mechanistic pathways linking obesity with COVID-19 symptom severity, in this review we first summarize clinical findings on obesity and COVID-19 disease outcomes.”

Comment #2:

Search strategy should be included.

Response: Thank you for pointing this out. We now added our search strategy in section 2.2, line 81-85: “We searched the Pubmed database up to March 15, 2021. In our search strategy, the combination of the following keywords was used: Obesity OR BMI OR overweight OR adiposity OR adipose tissue AND COVID-19 OR SARS-CoV-2. Our search was limited by published full-text article in English language” and in section 3 (line 123-126): “In this overview we discuss, in vitro, animal and human in vivo studies, including clinical trials. We searched the Pubmed database up to March 15, 2021. Our search was limited by published full-text article in English language.”

Comment #3:

Obesity has detrimental effects on respiratory mechanics, furthermore it may increase the risk of many metabolic and chronic disease (diabetes, cardiovascular disease, cancer...) that may co-exist in a single individual. These aspects may further influence mechanisms involved in covid-19 disease severity. Please add this aspect in the paper.

Response: We agree and mention these aspects at the beginning of section 3 (line 116-123): “To gain a better understanding of the pathophysiology of COVID-19 in patients suffering from obesity, in the next sections and in Figure 1 we provide an overview of the mechanistic pathways linking obesity with COVID-19 disease severity, with a focus on the metabolic and immunological consequences of obesity on COVID-19 disease course. Other features of obesity, like impaired respiratory mechanics and pulmonary function and the co-existence of metabolic disorders like diabetes and cardiovascular disease within a single individual also increase the risk for severe COVID-19 and complications [48],but are beyond the scope of this review.”

We also mention in the introduction section that the focus of this review will be on the metabolic and immunological consequences of obesity on COVID-19 disease course (line 42-44): “Here we focus on the metabolic- and immune-related consequences of obesity on COVID-19 disease course.” 
